# Lipid Droplet Accumulation Independently Predicts Poor Clinical Prognosis in High-Grade Serous Ovarian Carcinoma

**DOI:** 10.3390/cancers13205251

**Published:** 2021-10-19

**Authors:** Naoyuki Iwahashi, Midori Ikezaki, Masakazu Fujimoto, Yoshihiro Komohara, Yukio Fujiwara, Madoka Yamamoto, Mika Mizoguchi, Kentaro Matsubara, Yudai Watanabe, Ibu Matsuzaki, Shin-ichi Murata, Yoshito Ihara, Kazuhiko Ino, Kazuchika Nishitsuji

**Affiliations:** 1Department of Obstetrics and Gynecology, Wakayama Medical University, Wakayama 641-8509, Japan; naoyuki@wakayama-med.ac.jp (N.I.); madoka-y@wakayama-med.ac.jp (M.Y.); ma-mika@wakayama-med.ac.jp (M.M.); kazuino@wakayama-med.ac.jp (K.I.); 2Department of Biochemistry, Wakayama Medical University, Wakayama 641-8509, Japan; ikezaki@wakayama-med.ac.jp (M.I.); d1766082@wakayama-med.ac.jp (K.M.); d1766099@wakayama-med.ac.jp (Y.W.); y-ihara@wakayama-med.ac.jp (Y.I.); 3Department of Diagnostic Pathology, Kyoto University, Kyoto 606-8507, Japan; fujimasa@kuhp.kyoto-u.ac.jp; 4Department of Cell Pathology, Graduate School of Medical Sciences, Kumamoto University, Kumamoto 860-8556, Japan; ycomo@kumamoto-u.ac.jp (Y.K.); fuji-y@kumamoto-u.ac.jp (Y.F.); 5Department of Human Pathology, Wakayama Medical University, Wakayama 641-8510, Japan; m_ibu@wakayama-med.ac.jp (I.M.); smurata@wakayama-med.ac.jp (S.-i.M.)

**Keywords:** ovarian cancer, high-grade serous ovarian carcinoma, lipid droplet, adipophilin

## Abstract

**Simple Summary:**

High-grade serous carcinoma (HGSOC) is the most aggressive subtype of ovarian cancer and accounts for the vast majority of advanced stage cases. Intracellular accumulation of lipids as lipid droplets has been recognized as one of the characteristics of cancers and implicated in poor prognosis of several cancers, such as human melanomas. Here, we investigated the relationship between prognosis and lipid accumulation in HGSOC, and found that enhanced lipid accumulation in HGSOC tissues significantly correlated with poor prognosis. In cell-based assays with human ovarian cancer cells, we provide evidence that aerobic glycolysis, which is one of the characteristic metabolic abnormalities in cancer, induced lipid accumulation within cancer cells and targeting the lipid accumulation could suppress cancer cell proliferation. Thus, our results propose abnormal lipid accumulation as a negative indicator of HGSOC prognosis and a novel therapeutic target.

**Abstract:**

High-grade serous ovarian carcinoma (HGSOC) is an epithelial cancer that accounts for most ovarian cancer deaths. Metabolic abnormalities such as extensive aerobic glycolysis and aberrant lipid metabolism are well-known characteristics of cancer cells. Indeed, accumulation of lipid droplets (LDs) in certain types of malignant tumors has been known for more than 50 years. Here, we investigated the correlation between LD accumulation and clinical prognosis. In 96 HGSOC patients, we found that high expression of the LD marker adipophilin was associated with poor progression-free and overall survival (*p* = 0.0022 and *p* = 0.014, respectively). OVCAR-3 ovarian carcinoma cells accumulated LDs in a glucose-dependent manner, which suggested the involvement of aerobic glycolysis and subsequently enhanced lipogenesis, with a result being LD accumulation. The acyl-CoA: cholesterol acyltransferase 1 inhibitor K604 and the hydroxymethylglutaryl-CoA reductase inhibitor pitavastatin blocked LD accumulation in OVCAR-3 cells and reduced phosphorylation of the survival-related kinases Akt and ERK1/2, both of which have been implicated in malignancy. Our cell-based assays thus suggested that enhanced aerobic glycolysis resulted in LD accumulation and activation of survival-related kinases. Overall, our results support the idea that cancers with lipogenic phenotypes are associated with poor clinical prognosis, and we suggest that adipophilin may serve as an independent indicator of a poor prognosis in HGSOC.

## 1. Introduction

Metabolic reprogramming of tumor cells, including increased glucose uptake and upregulation of lipid metabolism, is a hallmark of cancer [1]. Lipid accumulation or the presence of abundant intracellular lipid droplets (LDs) in malignant tumors has been known for many years [2,3,4,5]. Several studies have confirmed the role of aberrant LD formation in promoting tumor cell survival in clear-cell renal cell carcinoma [6,7]. Assessment of LD formation in formalin-fixed paraffin-embedded sections has been difficult because of deparaffinization used in conventional immunohistochemical analysis. Recently, we and others successfully analyzed LD accumulation in various cancer tissues by using adipophilin (ADP) as a surrogate marker for LDs [8,9,10,11,12,13]. ADP, a member of the PAT (perilipin/ADRP/TIP47) family of proteins that coat LDs [14], has been proposed to produce LDs and increase the lipid load in non-adipose tissues under normal and disease conditions [15]. In our previous study, ADP accumulation significantly correlated with poor clinical prognosis and tumor cell proliferation in cutaneous malignant melanoma [13].

High-grade serous ovarian carcinoma (HGSOC) is the most lethal gynecologic malignancy and accounts for almost 80% of ovarian cancer mortalities [16,17,18]. More than 90% of these carcinomas have a mutation (or mutations) in the *TP53* gene, but they lack mutations in other genes that characterize low-grade serous carcinomas such as those in *KRAS*, *BRAF*, and *ERBB2* [19,20,21]. Thus, because of a lack of candidate genes for early diagnosis or efficient therapeutic targets, an increasing need exists for new strategies for early diagnosis and treatment of HGSOC. Here, we studied the correlation between LD accumulation and survival data for patients with HGSOC. In cell-based assays with an ovarian cancer cell line, enhanced LD formation closely correlated with activation of survival pathways. Our study will therefore improve the understanding of altered lipid metabolism in HGSOC.

## 2. Materials and Methods

### 2.1. Patients with HGSOC and Samples

We conducted a retrospective study of 96 female patients who had been newly diagnosed with ovarian cancer and who had undergone primary surgery or neoadjuvant chemotherapy (NAC) between January 2015 and March 2020 at Wakayama Medical University Hospital, Japan. Follow-up data and clinical information for all patients were obtained from patients’ charts. The senior pathologists M.F. and Y.K. examined hematoxylin and eosin-stained sections and provided histological diagnoses according to the criteria of the World Health Organization. Tumors were staged according to the International Federation of Gynecology and Obstetrics (FIGO) classification. Clinical findings related to images obtained from computed tomography, magnetic resonance imaging, and positron emission tomography/computed tomography, together with findings related to ascitic or pleural fluid cytology, were used to initially diagnose and stage patients who had NAC. For most patients who had had NAC, histological diagnoses were obtained after the post-NAC debulking surgery. Most of these patients, except those with stage IA disease, received 3−6 cycles of paclitaxel (175 mg/m^2^, day 1) and carboplatin (5 areas under the curve, day 1) with or without bevacizumab (15 mg/kg, day 1) every 21 days as postoperative therapy or NAC. Patients who manifested resistance to first-line chemotherapy received second-line regimens, which included mainly cisplatin (60 mg/m^2^, day 1) and irinotecan (60 mg/m^2^, days 1, 8, and 15). We excluded NAC patients whose cancer cells had marked or complete response to the chemotherapy. Table 1 provides a summary of the clinical data details. This study was approved by the ethics committee of Wakayama Medical University (authorization number: 1825) and was conducted in accordance with the tenets of the Declaration of Helsinki. All patients provided written informed consent for the use of their tissue samples.

### 2.2. Materials

OVCAR-3 human ovarian carcinoma cells, a HGSOC cell line [22], were provided by the RIKEN BioResource Research Center through the National BioResource Project of the Ministry of Education, Culture, Sports, Science and Technology /Japan Agency for Medical Research and Development, Japan. These cells were maintained in RPMI 1640 medium (Nacalai Tesque, Kyoto, Japan) supplemented with 10% heat-inactivated fetal bovine serum (Biowest, Nuaillé, France), 100 U/mL penicillin, and 100 µg/mL streptomycin (Wako Pure Chemical, Osaka, Japan) at 37 °C in a 5% CO_2_ atmosphere. Anti-β-actin antibody was purchased from Santa Cruz Biotechnology (Santa Cruz, CA, USA). Anti-ADP rabbit polyclonal antibody was obtained from LifeSpan Biosciences (Seattle, WA, USA). Rabbit polyclonal anti-phosphorylated Akt (S473) antibody, rabbit monoclonal anti-pan Akt antibody, and anti-pan extracellular signal-regulated kinase 1/2 (ERK1/2) antibody were obtained from Cell Signaling Technology (Beverly, MA, USA). Mouse monoclonal anti-di-phosphorylated ERK-1/2 was purchased from Sigma (St. Louis, MO, USA). The acyl-CoA-cholesterol acyltransferase 1 (ACAT-1) 1 inhibitor K604 and the hydroxymethylglutaryl-CoA reductase inhibitor pitavastatin were obtained from Sigma.

### 2.3. Immunohistochemical Analysis of ADP

Specimens were cut into 3-µm-thick sections. An anti-ADP antibody (Progen Biotechnik GmbH, Heidelberg, Germany) was used as the primary antibody. Samples were then incubated with horseradish peroxidase-labeled goat anti-mouse secondary antibody (Histofine; Nichirei Biosciences, Tokyo, Japan). Immunoreactions were visualized by using a diaminobenzidine substrate kit (Nichirei Biosciences). ADP expression was said to be positive when granular or vacuolar cytoplasmic expression was observed in the tumor cells. Foamy macrophages in the specimen were used as a built-in positive control, according to a previous study [11].

### 2.4. Scoring of ADP Immunostaining

The cytoplasmic area of each viable tumor cell that contained ADP-positive granules or vacuoles were calculated by means of a 0, 1+, 2+, and 3+ schema as previously described [13]. ADP immunostaining was quantified by using the H-score [(percentage at 1+) × 1+ (percentage at 2+) × 2+ (percentage at 3+) × 3].

### 2.5. Analysis of LD and ADP Expression

Cells were cultured in the presence of different concentrations of glucose (2.25, 4.5, or 9.0 g/L) for 72 h at 37 °C. To inhibit LD formation, cells were cultured in the presence of 9.0 g/L glucose and K604 (15 µM) or pitavastatin (5 µM). LDs were visualized by using the fluorescent probe Lipi-Green according to the manufacturer’s procedure (Dojindo, Kumamoto, Japan). ADP expression was analyzed by means of Western blotting and immunocytochemistry with an anti-ADP antibody. For Western blotting, OVCAR-3 cells were plated on 6-well culture plates and cultured as described above. Whole-cell lysates were prepared by means of the trichloroacetic acid precipitation method. Briefly, cells were washed three times with phosphate-buffered saline (PBS) and were then fixed with 10% (*w*/*v*) trichloroacetic acid in PBS on ice for 30 min, after which samples were centrifuged at 3000× *g* for 15 min at 4 °C. Precipitates were dissolved in sodium dodecyl sulfate-polyacrylamide gel electrophoresis sample buffer [0.125 M Tris-HCl; 4% (*w*/*v*) SDS, 20% (*v*/*v*) glycerol, 12% (*v*/*v*) 2-mercaptoethanol, and 0.01% (*w*/*v*) bromophenol blue]. Lysates were subjected to sodium dodecyl sulfate-polyacrylamide gel electrophoresis with 5–20% gradient gels and were then transferred to polyvinylidene difluoride membranes (Millipore, Billerica, MA, USA). ADP on the membranes was probed with an anti-ADP antibody (1:1000) followed by use of a horseradish peroxidase-labeled anti-rabbit antibody (1:10,000; Cell Signaling Technology) and ImmunoStar LD (Wako Pure Chemical). Protein contents of cell lysates were normalized to β-actin expression levels. Signals were detected by using a WSE-6100 LuminoGraph I (ATTO Corporation, Tokyo, Japan). For immunocytochemical analysis, OVCAR-3 cells were plated on poly-l-lysine-coated cover glasses and cultured as described above. Cells were then fixed with 4% paraformaldehyde in PBS for 20 min at room temperature. After being washed three times with PBS, cells were blocked and permeabilized with Animal-Free Blocker (Vector Laboratories, Burlingame, CA, USA) containing 0.05% saponin for 20 min at room temperature. They were then incubated with an anti-ADP antibody (1:100) followed by a Cy3-conjugated secondary antibody (1:1000; Jackson Immunoresearch, West Grove, PA, USA). Specimens were mounted with Vibrance Antifade Mounting Medium with DAPI (Vector Laboratories) and were examined with an LSM700 microscope (Zeiss, Oberkochen, Germany).

### 2.6. Analysis of Cell Survival-Related Pathways and Cell Proliferation

Activation of cell survival-related pathways was analyzed by means of Western blotting with an anti-phosphorylated Akt antibody and an anti-di-phosphorylated ERK antibody. OVCAR-3 cells were cultured with different concentrations of glucose (2.25, 4.5, or 9.0 g/L) for 72 h 37 °C in the presence or absence of K604 (15 µM) or pitavastatin (5 µM). Whole-cell lysates were prepared and Western blotting was performed as described above. Protein contents of cell lysates were normalized to β-actin expression levels. Cell viability was determined by using the WST-1 cell proliferation assay (Roche, Basel, Switzerland) according to the manufacturer’s protocol.

### 2.7. Statistical Analyses

Statistical analyses were performed by using Prism software (GraphPad Software, Version 7.04, La Jolla, CA, USA) and JMP Pro (Statistical Discovery Software, Version 13.1.0, Cary, NC, USA). For analysis of clinical data, comparisons between two groups were performed with Fisher’s exact test or the Wilcoxon test to analyze categorical and continuous variables, respectively. The Kaplan–Meier method was used to evaluate progression-free and overall survival rates, and log-rank tests were used to compare the groups. The Cox proportional hazards model was used to examine the relationship between clinicopathological parameters and survival. The correlation between the FIGO stages and ADP *H*-scores was analyzed via Kruskal–Wallis test followed by Dunn’s test. In the cell-based assays, statistical comparisons were performed by means of the unpaired Student’s *t* test or one-way analysis of variance followed by Dunnett’s test or Bonferroni’s test. Survival curves were compared via the Kaplan–Meier method and log-rank test. Results were said to be significant when *p* values were less than 0.05.

## 3. Results

### 3.1. Cohort Characteristics

The present study includes 96 patients with HGSOC who were staged according to the FIGO classification, that is, stage I: 4.2% (4/96); stage II: 7.3% (7/96); stage III: 71.9% (69/96); and stage IV: 16.7% (16/96) (Table 1). Table 1 summarizes other clinicopathological characteristics.

**Table 1 cancers-13-05251-t001:** Clinicopathological characteristics of patients with HGSOC.

	All (*n* = 96)	ADP-Low (*n* = 61)	ADP-High (*n* = 35)	*p* Value
Age, years (mean ± SD)	61.8 ± 11.2	61.8 ± 11.3	61.7 ± 11.3	0.985
BMI, kg/m^2^ (mean ± SD)	22.6 ± 4.4	21.9 ± 3.2	23.7 ± 5.8	<0.001 *
FIGO stage, *n* (%)				<0.001 *
I	4 (4.2)	4 (6.6)	0 (0.0)	
II	7 (7.3)	6 (9.8)	1 (2.9)	
III	69 (71.9)	48 (78.7)	21 (60.0)	
IV	16 (16.7)	3 (4.9)	13 (37.1)	
Recurrence or progression, *n* (%)	68 (70.8)	36 (59.0)	32 (91.4)	<0.001 *
Platinum sensitive, *n* (%)	31 (45.6)	16 (44.4)	15 (46.9)	0.999
Platinum resistant, *n* (%)	37 (54.4)	20 (55.6)	17 (53.1)	
Treatment, *n* (%)				
Neoadjuvant chemotherapy	40 (41.7)	20 (32.8)	20 (57.1)	0.0309 *
“Surgery alone” and adjuvant chemotherapy	56 (58.3)	41 (67.2)	15 (42.9)	
Death at the observation time point, *n* (%)	37 (38.5)	17 (27.9)	20 (57.1)	0.008 *
PFS, days [median (range)]	413 (16–5055)	971 (17–5055)	388 (16–1266)	0.004 *
OS, days [median (range)]	776 (16–5055)	1250 (17–5055)	847 (16–3662)	0.061

*** Statistically significant. ADP adipophilin, BMI body mass index, PFS progression-free survival, OS overall survival.

### 3.2. Impact of ADP Accumulation on HGSOC Prognosis

Figure 1 provides representative images of high and low ADP expression. The ADP staining pattern was not uniform, and not all cancer cells within a single cancer tissue accumulated ADP. No significant cytomorphological difference existed between HGSOCs with high ADP expression and those with low ADP expression. No tumor had bubbly or foamy cytoplasm indicating sebaceous differentiation.

H-scores of ADP immunostaining ranged from 0 to 150 (25% quartile, 5; median, 15; 75% quartile, 60). H-scores of 15 or lower and those higher than 15 were regarded as low and high ADP expression, respectively. We previously reported that several patients with cancers with high ADP expression had a poor clinical prognosis [11,13]. Here, we investigated the association of ADP expression and clinical prognosis in HGSOC. The mean age of all patients was 61.8 ± 11.2 years, and no significant difference was noted between the mean ages of patients with high ADP expression and those with low ADP expression. The median progression-free survival period for all patients was 413 days, with a range of from 16 to 5055 days; that for patients with low ADP expression was 971 days, with a range of from 17 to 5055 days, whereas that for patients with high ADP expression was 388 days, with a range of from 16 to 1266 days. The mean overall survival periods for all patients, patients with low ADP expression, and patients with high ADP expression were 776 days (range, 16–5055 days), 1250 days (range, 17–5055 days), and 847 days (range, 16–3662 days), respectively. Median H-scores of ADP immunostaining for patients at FIGO stage I, stage II, stage III, and stage IV were 3 (n = 4; range 0–20), 5 (n = 7; range, 0–60), 15 (n = 69; range 0–130), and 70 (n = 16; range, 0–150), respectively. H-scores of ADP immunohistochemistry correlated significantly with advanced the FIGO stages of the patients (Figure 2A). Kaplan–Meier analysis revealed that patients with high ADP expression had worse progression-free survival and overall survival rates compared with patients with low ADP expression (Figure 2B, C, *p* = 0.0022 and *p* = 0.0144, respectively). To clarify whether ADP immunostaining may serve an independent prognostic factor, we performed multivariate analyses. The multivariate Cox regression analysis demonstrated that ADP immunostaining (hazard ratio = 1.96; *p* = 0.0098) and the FIGO stage (hazard ratio = 5.58; *p* = 0.0017) were independent prognostic factors for impaired PFS, while ADP immunostaining was the only independent prognostic factor for impaired OS (hazard ratio = 2.18; *p* = 0.0245, Table 2). Use of NAC had no significant correlation with PFS and OS. These results clearly indicate that high ADP expression is a negative indicator of prognosis in HGSOC.

### 3.3. LD Formation and ADP Expression in OVCAR-3 Cells

One metabolic characteristic of cancer cells is related to the shift from oxidative phosphorylation as the major glucose consumer to aerobic glycolysis, also known as the Warburg effect [23]. Metabolic reprogramming in cancer cells includes a marked increase in de novo lipid synthesis in addition to aerobic glycolysis [24]. In our previous study, accumulation of ADP in human malignant melanoma cell lines depended on glucose concentration in the culture medium [13], which suggested a potential link between increased glucose consumption and lipid storage in malignant melanoma. Here, we used OVCAR-3 cells as a model of ovarian cancer. Formation of LDs in OVCAR-3 cells, as analyzed by means of Lipi-Green [25], depended on glucose concentration in the culture medium (Figure 3A). No signals were observed without Lipi-Green, which excluded the possibility of non-specific signals. Immunofluorescence analysis with an anti-ADP antibody revealed that ADP accumulation also depended on glucose concentration (Figure 3B), which supported the hypothesis that ADP immunostaining may be used as an indication of LD accumulation in cancer cells. We noted no cytotoxicity when OVCAR-3 cells were cultured with 2.25 g/L, 4.5 g/L, or 9.0 g/L glucose (Appendix A). In order to confirm that high glucose induced lipid droplet accumulation in HGSOC, we used an additional HGSOC cell line, OVKATE cells [22,26]. Lipid droplet formation and ADP expression in OVKATE cells depended on glucose concentration (Appendix A).

### 3.4. Glucose-Dependent Activation of Survival-Related Pathways in OVCAR-3 Cells

Dysregulation of several survival-related pathways, including the Akt and ERK1/2 pathways, has been implicated in the development and progression of cancers [27,28]. Because cellular lipid content may contribute to aberrant activation of these pathways [29,30], we next investigated whether glucose levels in culture media affected activation of Akt or ERK1/2. By using Western blotting with an anti-ADP antibody, LD formation in the presence of 9.0 g/L glucose was approximately 50% enhanced compared to that in the presence of 2.25 or 4.5 g/L glucose (Figure 4). We also found that levels of phosphorylated Akt and ERK1/2 increased with increasing glucose concentrations in the culture media (Figure 4). Levels of phosphorylated of Akt and ERK1/2 at 9.0 g/L glucose showed 50% to 100% increases compared to those at 2.2.5 g/L glucose. These results indicate that increased glucose consumption and the Warburg effect may lead to enhanced lipid storage and aberrant activation of survival-related kinases.

### 3.5. LD-Dependent Activation of Survival-Related Pathways

Cancer cells can enhance lipogenesis and cholesterol biogenesis [24,31]. We thus studied whether LD accumulation contributed to enhanced phosphorylation of the Akt and ERK1/2 pathways. For our studies here, we used K604, a selective inhibitor of ACAT-1 [32], that we previously reported suppressed the proliferation of glioblastoma cells [33]. ACAT-1 catalyzes the formation of cholesteryl esters, which are stored in LDs. Because perturbed cholesterol homeostasis is a well-known characteristic of cancer cells [34], we also used the cholesterol-lowering agent pitavastatin [35]. K604 and pitavastatin both efficiently prevented LD accumulation in OVCAR-3 cells in the presence of a high glucose concentration, as assessed by using Lipi-Green and ADP immunostaining (Figure 5A,B). We used K604 at 15 µM, because the treatment of OVCAR-3 cells with 10 µM K604 did not suppress ADP expression (Appendix A). Results with other ACAT inhibitors, TMP-153 [36,37] and Sandoz 58-035 [38], supported the involvement of ACAT in the LD accumulation (Appendix A). Amidepsine A, an inhibitor of acyl CoA:diacylglycerol acyltransferase (DGAT) [39,40], did not affect the LD and ADP accumulation, which may rule out the possibility that the LDs were made of triglycerides (Appendix A). Treatment of OVCAR-3 cells with K604 (15 µM) or pitavastatin (5 µM) significantly suppressed activation of the Akt (25% suppression by K604 and 40% suppression by pitavastatin) and ERK1/2 (12% suppression by K604 or pitavastatin) pathways (Figure 5C), which suggested that aberrant activation of the survival-related pathways at least partly depends on enhanced LD formation related to increased glucose consumption. Accordingly, K604 suppressed the proliferation of OVCAR-3 cells by approximately 20% (Figure 5D).

## 4. Discussion

Perturbed lipid metabolism has been recognized as a significant feature of malignant tumors [34]. To fuel the rapid growth of malignant cells, high rates of biosynthesis of macromolecules are required, the result being increased lipid synthesis as a common characteristic of human cancers [41]. Cancer cells thus need reservoirs for storing newly synthesized lipids, and they then can provide lipids for hydrolysis to supply fatty acids and cholesterol. In the present study, we showed a correlation between lipid accumulation and poor prognosis in HGSOC, which agrees with a previous suggestion that the malignant process of ovarian cancer causes metabolic changes such as those in glucose, amino acid, and lipid metabolism [42]. In our cell-based assays, LD accumulation depended on glucose concentrations. The Warburg effect, also known as aerobic glycolysis, is a metabolic hallmark of cancers, which are characterized by increased uptake and fermentation of glucose despite a sufficient supply of oxygen [23]. Increases in glucose uptake enable cancer cells to synthesize higher amounts of reducing equivalents, which eventually allow greater biomass synthesis such as de novo lipid synthesis [43,44]. Increased lipid synthesis frequently accompanies cancer progression and confers great advantages to cancer cells by providing lipids for rapid cell proliferation. Fatty acids are essential for progression of human cancers. Because high levels of glycolysis can provide cancer cells with energy and fatty acid precursors, one major consequence of the Warburg effect is increased de novo synthesis of fatty acids [45,46]. Esterified fatty acids in tumor cells have been thought to be derived from de novo synthesis [47]. Thus, results of our cell-based assay showing a correlation between increased glucose concentrations and LD accumulation suggest a common lipogenic phenotype of OVCAR-3 cells. Fatty acids in cancer cells can be esterified to form cholesteryl esters, which are stored in LDs. Cholesteryl esters in LDs can be hydrolyzed to produce fatty acids and cholesterol and thereby reduce the dependence of cancer cells on de novo fatty acid synthesis and avoid the energy costs of that synthesis. This lipogenic and lipid storage phenotype will thus offer advantages to cancer cells so that they survive insults such as low oxygen levels and lack of nutrients, which may provide an explanation for our observation that patients with HGSOC who have more LDs demonstrated poor clinical survival. However, currently it has not been demonstrated that cholesteryl esters can be a significant source of free fatty acids so that they fuel starving cancer cells. Elucidation of roles of cholesteryl ester-derived fatty acids in cancer cell growth deserves further studies. It is of note that BMIs significantly correlated with ADP expression in Table 1. The mechanism of how high BMI contributed to enhanced LD formation is not clear. Several studies suggested that serum levels of a certain adipokine, apelin, were increased in endometrial cancer patients with high BMIs compared to those with normal BMIs [48], and apelin was upregulated in tumor tissues of HGSOC patients with high BMIs [48]. In that study, BMI was related to both survival outcome and apelin immunoreactivity of HGSOC patients. Elucidation of the relationship between apelin and altered lipid metabolisms such as abnormal LD accumulation will be worth pursuing.

Although delipidation during deparaffinization processes makes it difficult to analyze detailed lipid compositions, our immunohistochemical analysis did in fact clarify whether LDs in patients with HGSOC stored triglycerides or cholesteryl esters. In our cell-based assays, inhibiting cholesterol esterification with the ACAT-1 inhibitor K604 efficiently suppressed LD accumulation in OVCAR-3 cells, which suggested that these cells stored cholesteryl esters in their LDs. Results with pitavastatin, which inhibits the rate-limiting enzyme of cholesterol synthesis, and amidepsine A, a DGAT inhibitor, supported this finding. Thus, accumulation of LDs in OVCAR-3 cells in the presence of a high glucose concentration may reflect an enhanced lipogenic phenotype related to the Warburg effect. Lipidomic analysis of frozen HGSOC tissues should aid the understanding of cancer-specific lipid metabolism.

In the present study, activation of the Akt pathway and ERK1/2 signaling depended on glucose concentrations in the culture media. Aberrant activation of the Akt signaling pathway has occurred in many cancers including HGSOC and has been implicated in HGSOC progression [27,49,50]. Levels of expression of phosphorylated Akt and the upstream phosphatidylinositol-4,5-bisphosphate 3-kinase catalytic subunit alpha were reportedly associated with survival in ovarian cancer, and the activation status of the Akt pathway has been suggested as an independent marker of prognosis in ovarian cancer [51]. Akt is reportedly cholesterol-sensitive [52], and at least a subpopulation of Akt is localized in the cholesterol-enriched lipid microdomains and activation of Akt depends on the microdomains [52,53]. Here, we showed that high glucose enhanced LD formation in ovarian cancer cells. Thus, it might be possible that the enhanced LD formation by high glucose altered the lipid composition of the plasma membrane, which may contribute to the activation of Akt. The ERK pathway plays a critical role in cell proliferation and cell survival [28]. In fact, the ERK pathway was reportedly activated in several cancer cell lines including ovarian cancer cell lines [54], and it has a major role in the pathogenesis of ovarian cancer [55]. Although de novo lipid synthesis does not occur in normal tissue cells, tumorigenesis is associated with a marked increase in lipid production [46,56]. In our study here, we showed that an inhibitor of cholesterol synthesis or esterification inhibited LD formation in OVCAR-3 cells that were cultured with a high glucose concentration. Inhibition of LD formation downregulated the activation of Akt and ERK1/2. These results strongly support the finding that LD accumulation that was induced by the Warburg effect reflects the lipogenic phenotype of OVCAR-3 cells and that the lipogenic phenotype plays a role in activation of survival-related signals. In addition to functioning as sites of lipid storage, LDs may also serve as sites of signal transduction of phosphatidylinositol 3-kinase, ERK1/2, p38, and PKCs, all of which have been implicated in oncogenic transformation, tumorigenesis, and metastasis [57,58,59]. Elucidating the detailed mechanisms of the oncogenic role of LDs will confirm the therapeutic importance of LDs and therapeutic efficacy of anti-lipogenic drugs in HGSOC. For now, we do not exclude the possibility that NAC may affect the LD formation in HGSOC patients (Table 1), and abnormal lipid metabolism including LD accumulation has been implicated in the chemoresistance in cancer [60]. Although our multivariate analysis clearly revealed that high ADP expression is an independent prognostic factor in HGSOC, investigation of whether LD accumulation may affect effectiveness of chemotherapy in HGSOC is also an important issue.

## 5. Conclusions

In summary, we found that LD accumulation in HGSOC was associated with poor prognosis and with activation of survival-related pathways, which suggested that high ADP expression may serve as a novel negative indicator of HGSOC prognosis. Our cell-based assays demonstrated that aerobic glycolysis may result in increased lipid storage and enhanced survival-related signaling, which confirm the correlation of ADP accumulation and poor prognosis in patients with HGSOC. The detailed mechanism by which lipids accumulated within cancer cells of these patients has not been clarified. Elucidating this detailed mechanism is necessary for understanding lipid metabolism in cancers and for providing a therapeutic target in HGSOC. Additional research is warranted to determine detailed mechanisms of how LD and ADP accumulation results in poor prognosis in HGSOC.

## Figures and Tables

**Figure 1 cancers-13-05251-f001:**
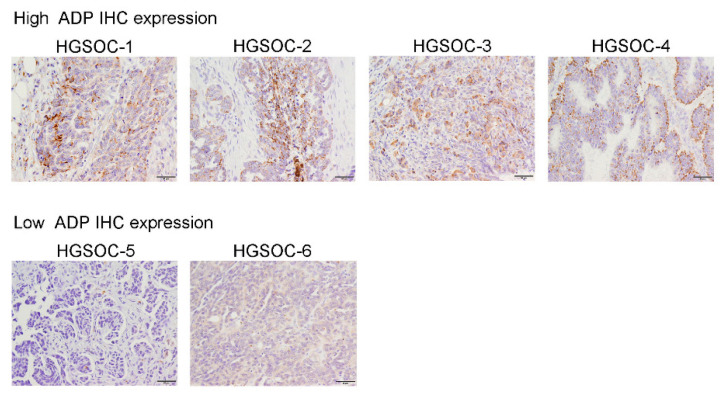
Immunohistochemical analysis of ADP in HGSOCs. Representative immunohistochemical (IHC) images of high and low ADP expression. In images with high ADP expression, ADP is expressed unevenly in the cytoplasm of tumor cells. The ADP staining pattern in predominantly granular with a few small vacuoles. No specific staining patterns were noted, but in a minority of cases ADP expression was enhanced in the subnuclear cytoplasm. Images with low ADP expression show that no significant cytomorphological difference existed compared with images with high ADP expression.

**Figure 2 cancers-13-05251-f002:**
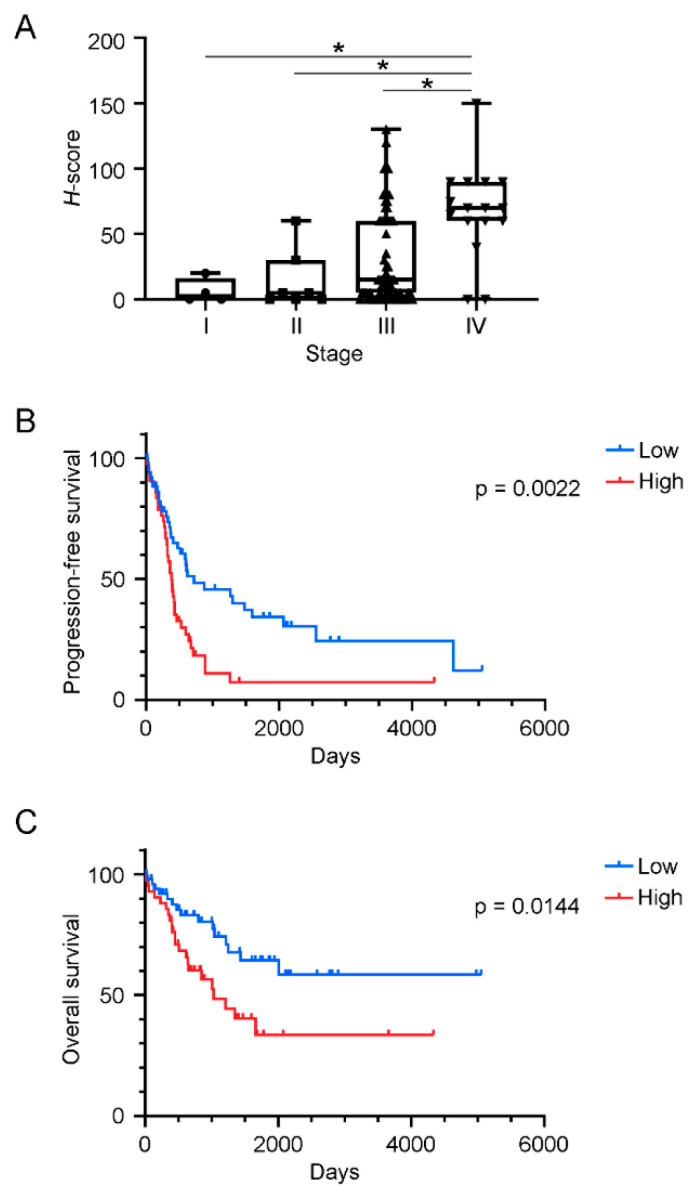
Correlation between clinical prognosis or clinical stages and immunohistochemical analysis of ADP in HGSOCs. (**A**) H-scores of ADP immunostaining for patients at different HGSOC disease stages. H-scores significantly increased according to disease stage. (**B,C**) Kaplan–Meier survival curves for progression-free survival rates (**B**) and overall survival rates (**C**) of patients with high ADP expression and patients with low ADP expression. * *p* < 0.05.

**Figure 3 cancers-13-05251-f003:**
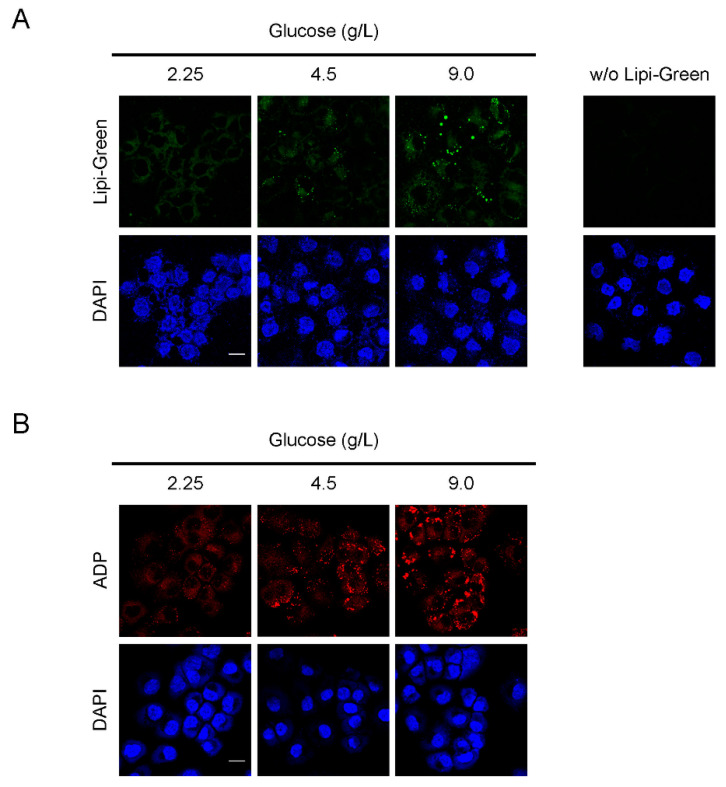
Glucose-dependent LD formation in OVCAR-3 cells. (**A**) OVCAR-3 cells were cultured with 2.25 g/L, 4.5 g/L, or 9.0 g/L glucose for 72 h, after which LDs in OVCAR-3 cells were visualized by using the lipophilic Lipi-Green. (**B**) After being cultured as described in (**A**), cells were fixed with 4% paraformaldehyde and immunostained with an anti-ADP antibody. Scale bars: 20 µm.

**Figure 4 cancers-13-05251-f004:**
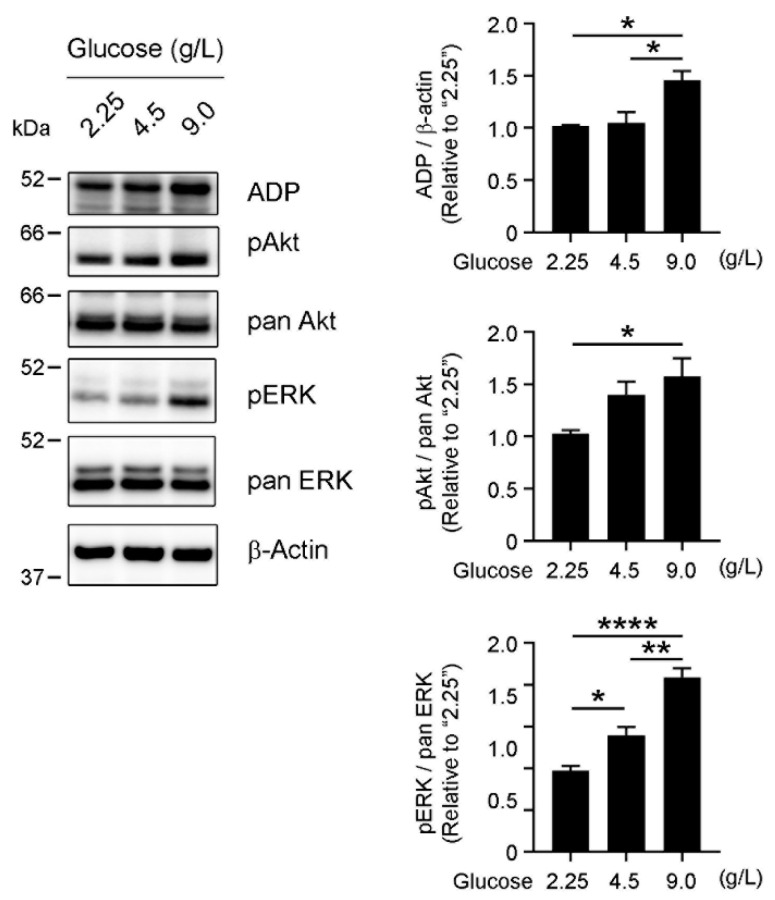
Glucose-dependent activation of survival-related kinase pathways. OVCAR-3 cells were cultured with 2.25 g/L, 4.5 g/L, or 9.0 g/L glucose for 72 h, after which ADP expression and phosphorylation of Akt and ERK1/2 were analyzed by means of Western blotting. ADP expression increased together with glucose levels in the culture media, with the greatest increase in LD formation in the presence of 9.0 g/L glucose. Phosphorylation of Akt and ERK1/2 was enhanced at higher glucose concentrations, which demonstrated glucose-dependent activation of these kinases. The graphs show quantifications of ADP, phosphorylated Akt, and phosphorylated ERK signals. Data are means ± S.E. of three independent experiments. β-Actin was used as a loading control. * *p* < 0.05, ** *p* < 0.01, **** *p* < 0.0001.

**Figure 5 cancers-13-05251-f005:**
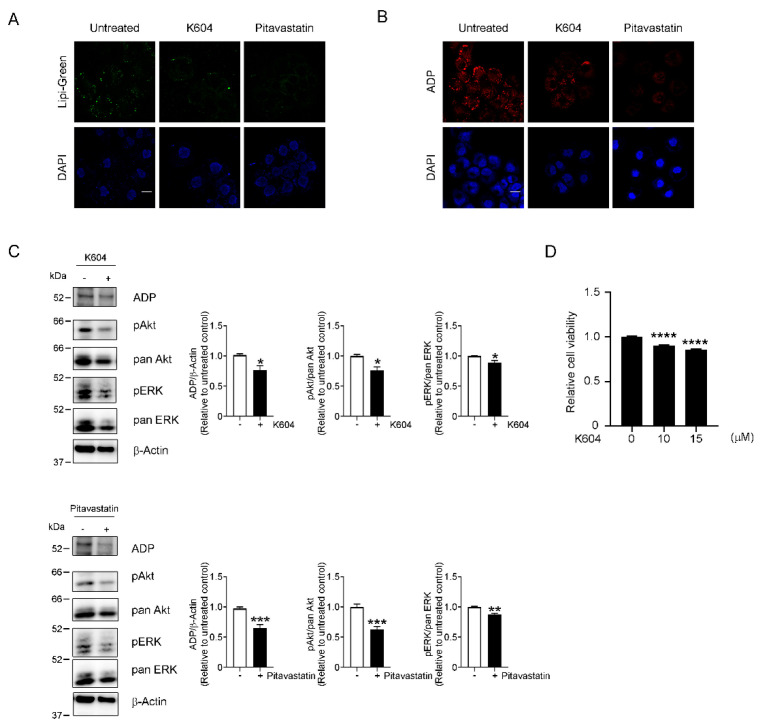
Requirement of LDs for activation of survival-related kinases. OVCAR-3 cells were cultured with 9.0 g/L glucose with or without K604, an ACAT-1 inhibitor (15 µM), or pitavastatin, a hydroxymethylglutaryl-CoA reductase inhibitor (5 µM) for 72 h, after which suppression of LD formation by these drugs was confirmed by using Lipi-Green (**A**) and ADP immunostaining (**B**). Scale bars: 20 µm. (**C**) OVCAR-3 cells were cultured as described above, and ADP expression and phosphorylation of Akt and ERK1/2 were analyzed by means of Western blotting. Inhibition of LD formation by K604 or pitavastatin suppressed activation of these kinases, which indicated involvement of LDs in activation of survival-related kinases. The graphs show quantifications of ADP, phosphorylated Akt, and phosphorylated ERK signals. Data are means ± S.E. of three independent experiments. β-Actin was used as a loading control. (**D**) OVCAR-3 cells were cultured as described above, and cell proliferation was analyzed by means of the WST-1 cell proliferation assay. Data are means ± S.E. of six independent experiments. * *p* < 0.05, ** *p* < 0.01, *** *p* < 0.001, **** *p* < 0.0001.

**Table 2 cancers-13-05251-t002:** Multivariate analysis of clinicopathological parameters.

Parameter	PFS Univariate		PFS Multivariate		OS Univariate		OS Multivariate	
	HR (95% CI)	*p* Value	HR (95% CI)	*p* Value	HR (95% CI)	*p*-Value	HR (95% CI)	*p* Value
Patient age	1.00 (0.98–1.02)	0.8154	1.00 (0.98–1.02)	0.9725	1.01 (0.98–1.04)	0.6564	1.01 (0.98–1.04)	0.5481
BMI	1.02 (0.96–1.07)	0.5684	1.00 (0.94–1.05)	0.9447	1.02 (0.95–1.09)	0.5134	1.00 (0.93–1.07)	0.9483
Stage (I-II vs. III-IV)	6.28 (1.54–25.70)	0.0005 *	5.58 (1.35–23.01)	0.0017 *	2.43 (0.58–10.09)	0.1630	1.98 (0.47–8.34)	0.3058
NAC (+ vs. −)	1.62 (0.99–2.62)	0.0523	1.06 (0.63–1.77)	0.8238	1.49 (0.78–2.84)	0.2289	1.09 (0.55–2.18)	0.8016
ADP (high vs. low)	2.15 (1.30–3.54)	0.0027 *	1.96 (1.17–3.27)	0.0098 *	2.24 (1.15–4.35)	0.0158 *	2.18 (1.10–4.34)	0.0245 *

* Statistically significant; ADP adipophilin, BMI body mass index, CI confidence interval, HR hazard ratio, OS overall survival, PFS progression-free survival.

## Data Availability

The data presented in this study are openly available in FigShare at https://doi.org/10.6084/m9.figshare.14691570.

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
