# Peer review of "Lipid Droplet Accumulation Independently Predicts Poor Clinical Prognosis in High-Grade Serous Ovarian Carcinoma"

_cancers, 2021, doi:10.3390/cancers13205251_

Round 1

Reviewer 1 Report

The authors addressed concerns partially, but the manuscript has been matured enough to be published. 

Reviewer 2 Report

The changes that you have made make the article of publishable quality.

This manuscript is a resubmission of an earlier submission. The following is a list of the peer review reports and author responses from that submission.

Round 1

Reviewer 1 Report

Comments-

  1. This paper covers a topic that is very intriguing, however, there are some additional experiments and changes required to strengthen claims made. The human cohort analysis believably shows that ADP levels correlate to disease progression and disease outcome. When looking at the cell culture model, it takes a rather high concentration of glucose to elicit the ADP response. Is 9g/L glucose physiologically justifiable in this type of cancer? Additionally, the concentration of K604 used throughout the paper is very high. The Ki of K604 is 0.45µM, 15µM seems unnecessarily high. K604 efficacy will vary slightly between cell lines, but maximal ACAT1 inhibition should occur at much lower concentrations. Testing the effects of K604 at lower concentrations is crucial to help rule out off-target effects. Additionally, the presumed effect of K604 is ACAT1 inhibition. Testing ACAT1 KD or KO would further rule out off-target effects and solidify its role in cancer cell survival.
  2. The authors stated that they showed via IHC that the lipid droplet accumulation in HGSOC patients were cholesteryl ester rich. This data was not shown in the figures provided. Regardless, it would be a good control to test DGAT inhibitors to rule out the role of triglycerides in the processes being studied.
  3. In order to provide a potential mechanism, the authors claim that, in cancer cells, cholesteryl ester hydrolysis releases free fatty acids which serve as a significant source of fuel for the cancer cells and that taking away lipid droplets and cholesteryl esters significantly starves the cancer cells of a fuel source. This is possible, but has it been demonstrated previously that cholesteryl ester hydrolysis provides a significant source of free fatty acids? Regardless, if cancer cells are in fact utilizing the fatty acids from cholesterol ester hydrolysis, simply adding exogenous fatty acids to cell media should be enough to negate the effects of K604.
  4. Finally, the authors referenced their previous work (reference 31) that investigated the effects of K604 on glioblastoma cells. That paper utilized a more appropriate concentration of K604. What is the explanation for increasing the concentration to 15µM here? Additionally, it would be good to look at the effects of K604 on a diseases relevant parameter like proliferation as was done previously. Interestingly, this previous work (reference 31) showed that lower concentration of K604 actually led to a reduction in ERK and AKT phosphorylation. Is there an explanation as to why K604 seems to have the opposite effect in OVCAR-3 cells?

Author Response

Reviewer 1

  1. This paper covers a topic that is very intriguing, however, there are some additional experiments and changes required to strengthen claims made. The human cohort analysis believably shows that ADP levels correlate to disease progression and disease outcome. When looking at the cell culture model, it takes a rather high concentration of glucose to elicit the ADP response. Is 9g/L glucose physiologically justifiable in this type of cancer? Additionally, the concentration of K604 used throughout the paper is very high. The Ki of K604 is 0.45µM, 15µM seems unnecessarily high. K604 efficacy will vary slightly between cell lines, but maximal ACAT1 inhibition should occur at much lower concentrations. Testing the effects of K604 at lower concentrations is crucial to help rule out off-target effects. Additionally, the presumed effect of K604 is ACAT1 inhibition. Testing ACAT1 KD or KO would further rule out off-target effects and solidify its role in cancer cell survival.

Reply: First, we would like to thank the reviewer for his/her comments to improve our manuscript. As for the glucose concentration, tumor cells consume a large amount of glucose. We hypothesized that high glucose consumption may lead to accumulation of lipid droplets and ADP, and the results with OVCAR-3 cells were supportive of that. In response to the reviewer’s concern, we performed a proliferation assay in the presence of various concentrations of glucose, and found that the glucose concentration of 9 g/L had no cytotoxicity (Supplemental Figure S1 and mentioned on page 8, lines 268–269). As for the concentration of K604, we agree with the reviewer that we used a higher concentration of K604 compared to the Ki against ACAT1, however, in this study, we analyzed the effect of K604 on lipid droplet formation and ADP accumulation, which are thought to occur as the consequence of the enzymatic function of ACAT. As shown in Supplemental Figure S3, 10 µM K604 failed to induce lipid droplet accumulation, and thus, we used K604 at 15 µM, which is also mentioned on page 10, lines 317–319. We have done ACAT1 KD with two siRNA (Silencer select validated siRNA SOAT1 (ACAT1), Ambion, # 4427038 and s13264; and SOAT1 (ACAT1) siRNA (h), Santa Cruz, #sc-29624), however, we failed to efficiently knock down ACAT1. Thus, we used another ACAT inhibitor, TMP-153 and Sandoz 58-035. As shown in Supplemental Figure S4A, B and on page 10, line 319–page 11, line 321, these inhibitors significantly suppressed the lipid droplet and ADP accumulation in OVCAR-3 cells, which supported the involvement of ACAT in the lipid droplet and ADP accumulation in the presence of high glucose.

  1. The authors stated that they showed via IHC that the lipid droplet accumulation in HGSOC patients were cholesteryl ester rich. This data was not shown in the figures provided. Regardless, it would be a good control to test DGAT inhibitors to rule out the role of triglycerides in the processes being studied.

Reply: We thank the reviewer for the constructive comment. We investigated the effect of a DGAT inhibitor, amidepsine A, on the lipid droplet accumulation in OVCAR-3 cells, and found that the DGAT inhibitor had no effect on the lipid droplet accumulation (Supplemental Fig. S4C and mentioned on page 11, lines 321–324, and on page 12, line 395).

  1. In order to provide a potential mechanism, the authors claim that, in cancer cells, cholesteryl ester hydrolysis releases free fatty acids which serve as a significant source of fuel for the cancer cells and that taking away lipid droplets and cholesteryl esters significantly starves the cancer cells of a fuel source. This is possible, but has it been demonstrated previously that cholesteryl ester hydrolysis provides a significant source of free fatty acids? Regardless, if cancer cells are in fact utilizing the fatty acids from cholesterol ester hydrolysis, simply adding exogenous fatty acids to cell media should be enough to negate the effects of K604.

Reply: To our knowledge, it has not been demonstrated that cholesteryl esters can be a significant source of free fatty acids. We have done experiments in which OVCAR-3 cells were supplied with exogenous free fatty acids, palmitic acid, however, exogenous palmitic acid failed to rescue the cytotoxicity of K604, possibly because of the cytotoxicity of exogenously-added palmitic acid itself such as inducing ER stress. Please see the below data for review process. It may be possible that exogenously-added fatty acids are well-known ER stress inducers, but intracellularly-generated fatty acids are not. We think that this is an important point and discussed on page 12, lines 378–381.

Review process Figure

OVCAR-3 cells were treated with K604 (15 µM) and palmitic acid (25 µM) for 72 h, after which cell viability was analyzed by means of the WST assay. Data are means ± SE (n = 8). ****, P < 0.0001.

  1. Finally, the authors referenced their previous work (reference 31) that investigated the effects of K604 on glioblastoma cells. That paper utilized a more appropriate concentration of K604. What is the explanation for increasing the concentration to 15µM here? Additionally, it would be good to look at the effects of K604 on a diseases relevant parameter like proliferation as was done previously. Interestingly, this previous work (reference 31) showed that lower concentration of K604 actually led to a reduction in ERK and AKT phosphorylation. Is there an explanation as to why K604 seems to have the opposite effect in OVCAR-3 cells?

Reply: In the present study, we investigated the relationship between lipid droplet formation and activation of the survival-related pathways. Because, in Figure 5A and Supplemental Figure S3, K604 at 15 µM, but not lower concentration 10 µM suppressed the lipid droplet formation, we used the concentration of 15 µM for further experiments. We noted this point on 10, line 317–319. In our previous study, we only focused on the effect of K604 on the cell proliferation, but not on lipid droplet accumulation. Furthermore, we used a glioblastoma cell line in that study, which is different from the cell lines in the present study. The effective concentration of K604 for inhibiting lipid droplet formation would depend on the types of cells, and as noted above, lower concentration of K604 failed to inhibit lipid droplet formation in OVCAR-3 cells. Thus, we used K604 at 15 µM. As the reviewer’s suggestion, we performed a cell proliferation assay, and the results are included in Figure 5D, which showed that K604 inhibited the proliferation of OVCAR-3 cells. The result was mentioned on page 11, lines 328–330. Here, because we looked at lipid droplet accumulation and activation of the survival-related pathways in the present study, we found that K604 at 15 µM reduced the phosphorylation of ERK and Akt. Actually, we are afraid but we think that K604 reduced the phosphorylation of ERK and Akt in both studies, and the effect of K604 does not seem to be “opposite”.

Reviewer 2 Report

Dear Authors,

In the study from authors Naoyuki Iwahashi et al. entitled ‘Lipid droplet accumulation independently predicts poor clinical prognosis in high-grade serous ovarian carcinoma.’, the authors analyzed lipid droplets (LD) in HGSOC and relation of LD and clinical impact and showed that high ADP staining is related to poorer prognosis. To analyze this in vitro, the authors used an HGSOC cell line OVCAR-3 and showed that high glucose concentration activates Akt signaling, probably by ACAT-1. This is an interesting study and the results support the author’s conclusion well. However, I would like to point out some weak points for publication. Especially, in vitro experiments require improvement as pointed-out followings.            

  1. In figures 4 and 5; 2.25, 4.5, and 9.0g/L glucose are not physiologic glucose concentrations in blood (9.0 g/L is a lethal concentration!). Please re-consider the design of in vitro experiments.   
  2. In vitro analysis using only one cell line (OVCAR-3) is not sufficient to be concluded. Please use additional cell lines.
  3. How high glucose can activate Akt pathway? Further mechanical analysis or deeper discussion is preferable to be shown.

Yours sincerely,

Author Response

Reviewer 2

In the study from authors Naoyuki Iwahashi et al. entitled ‘Lipid droplet accumulation independently predicts poor clinical prognosis in high-grade serous ovarian carcinoma.’, the authors analyzed lipid droplets (LD) in HGSOC and relation of LD and clinical impact and showed that high ADP staining is related to poorer prognosis. To analyze this in vitro, the authors used an HGSOC cell line OVCAR-3 and showed that high glucose concentration activates Akt signaling, probably by ACAT-1. This is an interesting study and the results support the author’s conclusion well. However, I would like to point out some weak points for publication. Especially, in vitro experiments require improvement as pointed-out followings.

  1. In figures 4 and 5; 2.25, 4.5, and 9.0g/L glucose are not physiologic glucose concentrations in blood (9.0 g/L is a lethal concentration!). Please re-consider the design of in vitro experiments.

Reply: First, we sincerely appreciate the reviewer’s critical reading and insightful comments. We are pleased that the reviewer appreciated the importance of our findings. Generally, tumor cells consume a lot of glucose. In cell-based assays, many factors, such as containing FBS in culture media, are not generally “physiological”, and actually, glucose concentrations of 2.25 and 4.5 in culture media such as DMEM (but not in the blood) are generally used as “low-glucose” and “high-glucose”. Here, we cultured OVCAR-3 cells in the presence of various concentrations of glucose. One of the purposes of the experiments is to find a glucose concentration that most efficiently induce lipid droplet accumulation and ADP expression in OVCAR-3 cells, and we found that the glucose concentration of 9.0 g/L. Thus, we used this glucose concentration in the subsequent experiments. However, we understand the reviewer ‘s concern, and analyzed the cell proliferation in the presence of various concentrations of glucose. The result is shown in newly prepared Supplemental Figure S1. As the result shows, the cell proliferation was not reduced in the presence of 2.25, 4.5, and 9.0 g/L glucose, but markedly suppressed in glucose concentrations of 13.5 and 18 g/L. Accordingly, we concluded that the glucose concentration of 9.0 g/L was not detrimental (and lethal) to OVCAR-3 cells. We noted this point on page 8, lines 268–269.

  1. In vitro analysis using only one cell line (OVCAR-3) is not sufficient to be concluded. Please use additional cell lines.

Reply: We performed experiments with the OVKATE cells, an ovarian cancer cells with HGSOC features (Mitra et al., Gynecol Oncol. 2015 138(2):372-7; Domcke et al., Nat Commun. 2013 4:2126), and confirmed that lipid droplet accumulation and ADP expression was enhanced in the presence of high glucose. The results are shown in newly prepared Supplemental Figure S2 and mentioned on page 8, lines 269–272.

  1. How high glucose can activate Akt pathway? Further mechanical analysis or deeper discussion is preferable to be shown.

Reply: Currently, we do not have an answer for why high glucose activate the Akt pathway. At least a portion of the Akt protein reportedly localized in the lipid microdomains (Adam et al., Cancer Res. 2007 67(13):6238-46), it may be possible that high glucose-induced lipid droplet accumulation might altered the membrane lipid composition via altering lipid metabolism in OVCAR-3 cells, which led to the activation of Akt. We discussed this on page 13, lies 407–412.

Reviewer 3 Report

The authors have reported that higher expression of Adipophilin as a surrogate marker of lipid droplets in 96 formalin-fixed sections of HGOSCs was associated with poorer prognosis. They also show that LD formation in a HGSOC cell line (OVCAR) increases with increasing glucose levels in the culture medium, and that this may be dependent on the activation of survival pathways.

It would have been useful to state that OVCAR cells are a HGSOC line in the methodology.

High BMI was significantly associated with high ADP levels in table 1. Some comment on this is needed in the results narrative and discussion.

57% of histological sections from patients who had NAC showed high ADP levels compared to 40% of sections from patients who had not been exposed to chemotherapy. Removing patients who had surgery alone (which is an unusual situation) from the analysis may reveal a statistical significance. This may be an important detail that relates to future clinical decisions, and should be commented on.

Author Response

Reviewer 3

The authors have reported that higher expression of Adipophilin as a surrogate marker of lipid droplets in 96 formalin-fixed sections of HGOSCs was associated with poorer prognosis. They also show that LD formation in a HGSOC cell line (OVCAR) increases with increasing glucose levels in the culture medium, and that this may be dependent on the activation of survival pathways.

  1. It would have been useful to state that OVCAR cells are a HGSOC line in the methodology.

Reply: First of all, we would like to thank the reviewer for his/her comments to improve our manuscript. We mention that OVCAR-3 cells are a HGSOC cell line on page 3, lines 103.

  1. High BMI was significantly associated with high ADP levels in table 1. Some comment on this is needed in the results narrative and discussion.

Reply: This is an interesting point. Previous studies reported that the serum levels of an adipokine apelin were higher in obese endometrial cancer patients than those of patients with normal BMI, and apelin immunoreactivity is correlated with BMI in HGSOC (Altinkaya et al., J Obstet Gynaecol Res. 2015 41:294–300; Unal et al., Biotech Histochem. 2020 95(1):27-36). Thus, it may be possible that the increased adipokines in patients with high BMIs may be involved in the enhanced cancer proliferation, lipid droplet formation and ADP expression. We discussed this on page 12, lines 381–388.

57% of histological sections from patients who had NAC showed high ADP levels compared to 40% of sections from patients who had not been exposed to chemotherapy. Removing patients who had surgery alone (which is an unusual situation) from the analysis may reveal a statistical significance. This may be an important detail that relates to future clinical decisions, and should be commented on.

Reply: This is an important point. We re-analyzed the data and found that use of NAC is positively correlated with high ADP. Please see the revised Table 1. We also analyzed the association of NAC and PFS or OS, and found that NAC did not significantly correlate with PFS and OS. Please see the revised Table 2. Thus, it is possible that NAC might have affected ADP expression, however, our multivariate analysis revealed that high ADP expression was an independent prognostic factor of HGSOC. We mentioned and discussed these points on page 6, lines 227 and page 13, lines 427–432. 
